# Clinical and Psychosocial Characteristics of Adolescent Pediatric Patients Hospitalized after Different Types of Suicidal Behaviors—A Preliminary Study

**DOI:** 10.3390/ijerph17155568

**Published:** 2020-08-01

**Authors:** Bartosz Bohaterewicz, Magdalena Nowicka, Anna M. Sobczak, Aleksandra A. Plewka, Patrycia Gaszczyk, Tadeusz Marek

**Affiliations:** 1Department of Psychology of Individual Differences, Psychological Diagnosis, and Psychometrics, Institute of Psychology, University of Social Sciences and Humanities, 03-815 Warsaw, Poland; bohaterewicz@gmail.com (B.B.); nowickamagda@op.pl (M.N.); aleksandra.plewka@icloud.com (A.A.P.); 2Department of Cognitive Neuroscience and Neuroergonomics, Institute of Applied Psychology, Jagiellonian University, 31-007 Cracow, Poland; tademarek@gmail.com; 3Department of Psychology, University of Essex, Colchester CO4 3SQ, UK; gaszczykpati54@gmail.com

**Keywords:** adolescents, suicide, psychosis

## Abstract

The objective of this study was to examine the clinical characteristics of adolescents hospitalized after a suicide attempt or instrumental suicide-related behavior. Participants included thirty-six adolescents from the pediatric unit of a Polish hospital who made a nonfatal suicide attempt (SAA) or engaged in instrumental suicide-related behavior (IBA), as well as a general population sample (GPS). Psychosocial features were measured using the Kiddie Schedule for Affective Disorders and Schizophrenia (K-SADS), the Social and Occupational Functioning Assessment Scale (SOFAS), the Suicide Behaviors Questionnaire–Revised (SBQ-R), the Psychache Scale (TPS), the State–Trait Anxiety Inventory (STAI), the Center of Epidemiological Studies Depression Scale for Children (CES-DC), and the Prodromal Questionnaire (PQ-16). The SAA group scored significantly higher than the IBA group and the GPS in modules related to irritability and anhedonia, voice hallucinations and delusions, suicidal acts, thoughts and ideation, and medical lethality. Additionally, the SAA scored higher on the SBQ-R and PQ-16 compared to the IBA group and the GPS. Although anxiety, mental pain, and depressive symptoms could not independently distinguish between the SAA and IBA groups, psychotic symptoms were more frequently present within the SAA group. The above symptoms may be important to consider when screening for suicide risk in the general population.

## 1. Introduction

Suicide-related behaviors, including suicide, non-fatal suicide attempts, suicide ideation, and instrumental suicide-related behaviors, are a matter of great concern for mental health specialists since suicide is the third-leading cause of death among youth worldwide [1]. A recent community study in Poland revealed that 16% of adolescents have been engaged in some type of non-suicidal self-injury, whereas 7% of Polish youth had attempted suicide [2]. These rates indicate the importance of investigating the psychosocial profile of adolescents with different forms of suicide-related behaviors to prevent this public health problem. While literature reviews unanimously point to psychosocial risk factors in the occurrence of suicide-related behavior, biological factors, including brain development, should also be mentioned. Research on brain development and connectivity between its parts shows that in adolescents’ brains, regions of the amygdala and hippocampus develop earlier than the prefrontal cortex, which is responsible for impulse regulation [3]. This can be observed in frequent mood switches and impulsive behaviors among youth. Imbalances between brain regions’ maturity occur naturally in the brain development of this group; however, among those teenagers who are at risk of suicide-related behavior (SRB) or were involved in one, less synchronized activation between the amygdala and the medial prefrontal cortex can be observed. Non-synchronized activation of these two brain regions can act as a biomarker for increased vulnerability for potential mood disorders, and consequently, suicide-related behaviors [4].

Immediately after various SRBs, adolescents are typically pediatrically hospitalized in acute care hospitals [5]. Admissions for suicide attempts in USA children’s hospitals increased steadily from 2008 to 2015 and account for an increasing percentage of all hospital stays across all age groups [5]. Meanwhile, in Canada, Rhodes et al. [6] reported that up to 80% of all adolescents who attempt suicide visited a healthcare provider or an emergency room in the year before their death. In Poland, only a few studies addressed the characteristics of adolescents at risk of SRB, one of which was conducted on 1448 secondary school teenagers aged 12–19 years from one of the bigger Polish cities [7]. It is important to mention that even though this study covered a large sample of participants, it focused exclusively on skin-cutting behavior as a form of self-harm. Another study of Gmitrowicz et al. [8] took into consideration suicide ideation and suicide attempts; however, it did not cover instrumental SRBs.

The above data, combined with the fact that in 90% of cases, parents of adolescents are fully unaware of their children’s suicide ideations after a suicide attempt [9,10], highlights the increasing role pediatric hospital services play in treating adolescents with various SRBs. 

Among patients hospitalized at pediatric departments after SRBs, 1.9% of adolescents in the general population engage in instrumental suicide-related behaviors [11], which are:
potentially self-injurious behavior[s] for where there is evidence (either implicit or explicit) that (a) the person did not intend to kill him/herself, and (b) the person wished to use the appearance of intending to kill him/herself in order to attain some other end (e.g., to seek help, to punish others, to receive attention) [12].

Additionally, 2.7% of adolescents in the general population are hospitalized after suicide attempts [11], which are:
potentially self-injurious behaviors with a non-fatal outcome for which there is evidence (either explicit or implicit) that the person intended at some (non-zero) level to kill himself/herself. A suicide attempt may or may not result in injuries [12].


Although many risk factors of suicide have been identified in the adolescent population [13], no specific tests are individually capable of identifying a suicidal person with high certainty. This is complicated by the fact that there are various types of SRBs. To date, only a few studies have investigated the clinical characteristics of adolescent patients after different types of SRBs [14]. Based on existing data, we are not able to decide: (1) which factors predict the transition from suicidal ideation (SI) to a suicide attempt (SA), (2) which factors are particularly strong predictors of this transition, and (3) which factors best and most independently predict self-injurious behaviors. In summary, there are significant gaps in the knowledge regarding potential risk factors of different types of SRBs, which may be important in not only the identification of adolescents in danger of attempting suicide but also in providing these adolescents with the right intervention. 

### 1.1. Clinical Characteristics of Adolescents With SRBs

A vast majority of studies suggest that depressive symptoms are the most general predictor of SRBs [15,16,17,18,19,20]. However, some results suggest that there are differences in depression scores between adolescents who have performed suicidal versus non-suicidal self-harm [14,21]. Moreover, depressive disorders are consistently the most prevalent psychiatric disorder among adolescents who died by suicide (prevalence ranging from 49% to 68%) [22]. Therefore, it is still unclear how depressive symptoms are related to different types of SRBs in adolescent patients. Anxiety can also be treated as a causal risk factor for different forms of SRBs [23]. Biuckians, Miklowitz, and Kim [24] associated anxiety scores with suicide attempts and suicidal gestures, while Boden, Fergusson, and Horwood [25] showed that the presence of any anxiety disorder was associated with an increased risk of suicide ideations or suicide attempts among 16- to 18-year-olds. The aforementioned data support anxiety as an additional general predictor of SRBs in children and adolescents, but similarly to depression, it is unclear how anxiety is related to different types of SRBs. 

It seems that psychotic symptoms may more differentially relate to different types of SRBs. Research by Falcone et al. [26] has shown that children and adolescents with psychosis are more likely to have multiple repeated suicide attempts than non-psychotic psychiatric inpatients. Research has also found a high prevalence of suicide ideations among patients at ultra-high risk of psychosis or psychotic-like experiences [27,28]. Unfortunately, this potential difference in the presence of psychotic symptoms has not yet been demonstrated in population-based studies with a direct comparison of adolescent groups after different SRBs. 

Studies on SRBs often focus on mental pain [29,30,31]. Shneidman [29] was the first scientist who associated mental pain with suicide risk and claimed that suicide occurs when mental pain is deemed by the person to be unbearable. The systematic review conducted by Verrocchio et al. [32] found mental pain alone to be the predictive factor of suicide risk regardless of the diagnosed mental disorder. However, it remains unknown to what extent mental pain is related to different forms of SRBs in adolescents. 

Although numerous studies have identified psychopathological risk factors, surprisingly few have examined the relationship between social functioning and SRBs. Berman and Schwartz [33] reported that a third of adolescents who attempted suicide had shown interpersonal concerns before the attempt. Other studies suggest that various indices of social isolation, such as thwarted belongingness [34], living alone [35], loneliness [36], or low social support [37,38,39] are associated with an increased risk of suicide. 

### 1.2. Objectives

The literature on the psychosocial features of adolescents who have a history of different SRBs is limited and developing. The purpose of this study was to examine the psychosocial differences between adolescent pediatric patients hospitalized after suicide attempts and instrumental SRBs, as SRBs are common among adolescents with mental health problems. We concentrated on variables such as depression, anxiety, psychotic symptoms, mental pain, and the level of interpersonal functioning. We hypothesized that adolescent patients with a history of non-fatal suicide attempts present a more serious psychopathological picture than adolescent patients with a history of instrumental SRBs. 

## 2. Materials and Methods

### 2.1. Participants

Over 12 months, we identified 38 adolescents hospitalized for SRBs, representing 3.39% of the total number of pediatric patients (n = 914) of a district hospital in central Poland. The identified patients were divided into two groups: (1) fifteen non-fatal suicidal act adolescents (SAAs; 1.22% of the pediatric patient population) and (2) twenty-three instrumental behavior adolescents (IBAs; 2.16% of the pediatric population). The above classification was made by a clinical psychologist and a pediatrician, who evaluated the suicide-related behaviors presented by patients, the specificity of hospital care (length of stay, psychiatric consultations), and the follow-up course (coordination with the general practitioner and/or social services, recommended aftercare), based on the recommendations of O’Caroll et al. [10]. Eleven patients from the IBA group, along with three participants from the SAA group, did not agree to participate in the questionnaire part of the study. Additional exclusion criteria for the SAA and IBA groups were gross reality distortion and the need for inpatient psychiatric admission. Finally, questionnaire data from twelve SAA (aged 13–18 years old) patients and twelve IBA (aged 13–17 years old) patients were evaluated. The control group (general population sample; GPS) consisted of twelve adolescents recruited from secondary and middle schools (aged 12–16 years old) without psychiatric disorders, as assessed by a clinical psychologist. They were age- and gender-matched to the SAA and IBA groups.

In the SAA group, the average hospital stay was 8.2 days (SD = 7.9). In this group, life-saving medical procedures were used during their hospital stay (e.g., advanced cardiac life support, hemorrhage control, gastric lavage, pharmacological treatment). In the IBA group, the average hospital stay was 4.2 days (SD = 3.2). In this group, medical procedures were limited to diagnostics that helped measure the severity of the problem and/or basic pharmacological treatment (analgesics, anti-diarrhea drugs, laxatives).

During hospitalization, all patients from both clinical groups were consulted by a clinical psychologist and a psychiatrist. Both patient groups were diagnosed according to the International Statistical Classification of Diseases and Related Health Problems (ICD-10) criteria. The SAA group was diagnosed with the following mental disorders: depression (n = 2), anxiety disorder (n = 4), conduct disorder (n = 3), adjustment disorder (n = 2), schizoaffective disorder (n = 1), and personality disorder (n = 2). The IBA group was diagnosed with the following mental disorders: depression (n = 5), conduct disorder (n = 6), and anxiety disorder (n = 1). 

Written and informed consent was obtained from all parents of participants. Child participants provided verbal assent following Frost’s [40] guidelines on the ethical inclusion of children with psychiatric problems in research. The study was approved by the Research Ethics Committee of Faculty of Psychology at SWPS University of Social Sciences and Humanities (no. 23/2018). 

### 2.2. Measures

#### 2.2.1. Diagnostic and Statistical Manual of Mental Disorders (DSM-5) Diagnosis/Psychotic Symptoms

Clinical diagnoses were recorded using the DSM-5 criteria. The Polish version of the Kiddie Schedule for Affective Disorders and Schizophrenia (K-SADS) was used by clinicians during the diagnostic process [41]. The K-SADS is a semi-structured diagnostic interview designed to assess current and past episodes of psychopathology in children and adolescents. The suicide module of the K-SADS consists of five items: recurrent thoughts of death, suicidal ideation, suicidal acts—intent, suicidal acts—medical lethality, and non-suicidal self-injurious behavior. The depression module of the K-SADS consists of three items: depressive mood; irritability and anger; and anhedonia, lack of interest, apathy, low motivation, and boredom. The psychosis module of the K-SADS consists of three items: voice hallucinations, visual hallucinations, and delusions. Probes and objective criteria are provided to rate individual symptoms (scores between 0 and 3, where 0 means “no information” and 3 means “threshold”). Only 3-point scores were taken to signify present symptoms, indicating possible depression and anger problems, as well as a high risk of psychosis and suicide. K-SADS showed good overall convergent and discriminant validity and good inter-rater reliability [41]. 

#### 2.2.2. Depression

The Center of Epidemiological Studies Depression Scale for Children (CES-DC) was used to assess depression scores. CES-DS is a 20-item measure of self-reported symptoms experienced in the past week that are associated with depression, such as restless sleep, poor appetite, and feeling lonely [42]. Response options range from 0 to 3 for each item. Scores range from 0 to 60, with high scores indicating greater depressive symptoms. The CES-DC shows good psychometric properties in a population of adolescents (α = 0.84) [43]. 

#### 2.2.3. Anxiety

Data concerning the level of anxiety were collected using the State–Trait Anxiety Inventory (STAI). The STAI is based on a four-point Likert scale and consists of 40 self-report questions. It measures two types of anxiety: anxiety as a state (A-state) and anxiety as a trait (A-trait) [44]. Higher scores are positively correlated with higher levels of anxiety. The test–retest reliability for this inventory was found to be 0.97 for A-trait and 0.45 for A-state [45]. 

#### 2.2.4. Psychache

The Psychache Scale (TPS) is a self-assessment method that is designed to assess subjectively experienced psychological pain [46]. It consists of 13 statements scored between 1 and 5 points. The total score is a sum of all points, with a higher score meaning greater mental pain. Relative to other established psychological antecedents of suicide measures, TPS has strong internal consistency reliability (α = 0.92), criterion validity, and incremental validity for statistically predicting one’s status on these suicide criteria [46].

#### 2.2.5. Psychotic Symptoms

The Prodromal Questionnaire (PQ-16) was used as a 16-item self-reported screener in the early detection of psychosis risk and psychotic symptoms [47]. Nine items of this questionnaire assess perceptual abnormalities and hallucinations. Five items assess unusual thought content, as well as delusional ideas and paranoia. Two items assess negative symptoms. Scores are given between 0 and 3, where 0 indicates a lack of symptoms and 3 indicates the full presence of symptoms. In preliminary research, the PQ-16 showed satisfactory internal consistency (Cronbach’s α = 0.65) [48].

#### 2.2.6. Social Functioning

To assess the level of social functioning, we used the Social and Occupational Functioning Assessment Scale (SOFAS) [49]. SOFAS is clinically rated and assesses impairments in functioning due to physical limitations, as well as those due to mental impairments. The rating of overall psychological functioning is on a scale of 0–100. SOFAS has a suitable validity and reliability for the assessment of functioning in patients with different forms of psychopathology (kappa coefficient = 0.85) [50].

#### 2.2.7. Risk Factor for Suicide

For each participant, we identified risk factors for suicide using the Suicide Behaviors Questionnaire–Revised (SBQ-R), which is a self-report questionnaire that has four items, each assessing a different aspect of suicidality [51]. The first item concerns the presence of suicidal thoughts and attempts. The second item is about the frequency of suicidal thoughts. The third item estimates the threat level of suicidal attempts. The fourth assesses the likelihood of future suicidal attempts. Each question has an individual scale, and each response corresponds to a certain point value. The total score ranges from 3 to 18, with a higher score indicating a greater risk of suicide. The original version of the SBQ-R showed satisfactory internal consistency (Cronbach’s α = 0.88) [51].

### 2.3. Statistical Analyses

Group differences in non-normally distributed, ordinal-scaled data were analyzed using Kruskal–Wallis tests with post hoc pair-wise Mann–Whitney U-tests, alongside the exact significance option to account for the small sample size. Quantitative variables were analyzed using a one-way analysis of variance (ANOVA). Group differences in frequencies were calculated using χ^2^ tests and, when expected cell frequencies were below 5, using the Fisher’s exact test. Alongside the significance level, the effect size was reported as η^2^ for the Kruskal-Wallis, one-way ANOVA, and Mann–Whitney U-test. Statistical analyses were carried out using IBM SPSS Statistics for Windows, Version 24.0 (IBM Corp., Armonk, NY, USA). 

## 3. Results

Methods used by patients during SRBs were classified according to the international self-harm section of the ICD-10 codes: self-poisoning (X60–69), drowning (X70), firearms (X72–74), burning (X75–77), self-harm using a sharp object (X78), self-harm using a blunt object (X79), jumping from a high place (X80), and others (X81–84). Common SRBs were cutting and drug intoxication. Although medication overdose was the most frequent SRB in both hospitalized groups (over 80% of the IBA group and over 75% of the SAA group), we observed differences between groups in the specificity and quantity of ingested drugs. The participants of the SAA group who had decided to overdose medications usually chose medicines used for somatic ailments (67%), such as Paracetamolum, Helides, Thiocodin, or cardiac medication, in comparison to psychotropic medication (33%). Surprisingly, in the IBA group, psychotropic medications were much more common (70%) than somatic medications (30%). The SAA group overdosed more (M = 28 pills, SD = 4.2) than the IBA group (M = 8 pills, SD = 4.1). Drug overdose effects were evaluated by pediatricians as much more toxic in the SAA group (sedation, hyper-stimulation, a seizure emergency) than in the IBA group (abdominal pain, headache). The second-most frequent method of suicide in the SAA group was wrist cutting (8%). In the IBA group, self-cutting behaviors were more frequent (33%) and were less serious (wrist scratches). 

The Kruskal–Wallis test revealed no differences in gender, serious injuries, head injuries, or significant medical health problems between the groups. However, there were significant between-group differences in the number of previous hospitalizations (*p* < 0.001) and problems with social relatedness (*p* < 0.001), which were greater in the SAA group. The one-way ANOVA revealed no significant differences in age between the groups. The participant’s detailed demographics and clinical characteristics are displayed in Table 1.

Normally distributed and ratio-scaled data obtained from CES-DC, TPS, and STAI were compared using a one-way analysis of variance (ANOVA) with a Sidak post hoc test. For the CES-DC and TPS, two individuals were excluded from further analysis due to their scores being more than three standard deviations away from their group’s average score. There was a significant main effect of STAI (x-2) (ANOVA, F(2,33) = 32.128; *p* < 0.001) and STAI (x-1) subscales (ANOVA, F(2,33) = 34.634; *p* < 0.001), TPS (ANOVA, F(2,32) = 71.470, *p* < 0.001), and CES-DC (ANOVA, F(2,32) = 207.217; *p* < 0.001). 

The Sidak post hoc tests revealed:(a)The IBA and SAA groups had a significantly higher level of trait anxiety than the GPS group (M_IBA_ = 51.583, SD = 10.004 > M_GPS_ = 30.250, SD = 5.413, *p* < 0.001; M_SAA_ = 55.5, SD = 8.806 > M_GPS_ = 30.250, SD = 5.413, *p* < 0.001). No significant (*p* > 0.05) difference was found between the IBA and SAA group (*p* > 0.05). We also observed a significantly higher level of anxiety as a state in both the IBA and SAA groups compared to the GPS group (M_SAA_ = 60.67, SD = 9.06 > M_GPS_ = 35.250, SD = 7.021, *p* < 0.001; M_IBA_ = 56.75, SD = 7.96 > M_GPS_ = 35.250, SD = 7.021, *p* < 0.001). No significant differences were found between the IBA and SAA groups (*p* > 0.05).(b)Both the SAA and the IBA groups had significantly higher levels of depressive symptoms than the GPS group (M_SAA_ = 56.08, SD = 3.03 > M_GPS_ = 32.67, SD = 2.6, *p* < 0.001; M_IBA_ = 47.42, SD = 11.07 > M_GPS_ = 32.67, SD = 2.61; *p* < 0.001). Additionally, the SAA group had a significantly higher level of depressive symptoms compared to the IBA group (M_SAA_ = 56.08, SD = 3.03 > M_IBA_ = 47.42, SD = 11.07, *p* < 0.001).(c)The SAA and IBA groups were characterized by a significantly higher level of subjectively experienced psychological pain (*p* < 0.001) than the GPS group (M_SAA_ = 41.181, SD = 6.242 > M_GPS_ = 17.667, SD = 4.097; M_IBA_ = 36.083, SD = 4.542 > M_GPS_ = 17.667, SD = 4.097). There was no significant (*p* > 0.05) difference between the IBA and SAA groups.

The group means are displayed in Table 2.

A Kruskal–Wallis H test revealed a significant (*p* < 0.001) difference between the three groups in terms of the SBQ-R and PQ-16 scores. Non-normally distributed SBQ-R and PQ-16 scores were then compared using post hoc pair-wise Mann–Whitney U-tests. Scores from two individuals in the GPS group were excluded from further analysis because their data was more than three standard deviations from the mean. In the SAA group, we observed a significantly higher risk of suicide (Mdn_SAA_ = 18.5 > Mdn_ISB_ = 6.5), as well as significantly higher risk of psychosis (Mdn_SAA_ = 18.5 > Mdn_ISB_ = 6.5) than in the IBA group (Table 3). 

Ordinal-scaled data obtained from the K-SADS interview were compared using a Kruskal–Wallis test, which revealed significant differences (*p* < 0.05) in the presence of symptoms between the three groups. A Mann–Whitney U test revealed that the SAA group scored significantly higher than the IBA group in the irritability and anger module (Mdn_SAA_ = 14.792 > Mdn_IBA_ = 10.208); the recurrent thoughts of death module (Mdn_SAA_ = 17.167 > Mdn_IBA_ = 7.833); the suicidal ideation module (Mdn_SAA_ = 16.125 > Mdn_IBA_ = 8.875); the non-suicidal, self-injurious behavior module (Mdn_SAA_ = 15.583 > Mdn_IBA_ = 9.417); the suicidal acts—medical lethality module (Mdn_SAA_ = 15 > Mdn_IBA_ = 10); and the suicidal acts—medical lethality in the past module (Mdn_SAA_ = 14.455 > Mdn_IBA_ = 9.75). The IBA group scored higher than the GPS group in the suicidal acts module (Mdn_IBA_ = 17.833 > Mdn_GPS_ = 7.167); the suicidal ideation module (Mdn_SAA_ = 16.083 > Mdn_GPS_ = 8.917); the suicidal acts—medical lethality module (Mdn_IBA_ = 18.5 > Mdn_GPS_ = 6.5); the suicidal acts—medical lethality in the past module (Mdn_IBA_ = 9.5 > Mdn_GPS_ = 15.5); and the non-suicidal, self-injurious behavior module (Mdn_IBA_ = 14.5 > Mdn_GPS_ = 10.5). The SAA group reported significantly more positive symptoms than the IBA group (hallucinations—voices module, Mdn_SAA_ = 17 > Mdn_IBA_ = 8; delusions module, Mdn_SAA_ = 16.04 > Mdn_IBA_ = 8.96). Moreover, risk of suicide (Mdn_IBA_ = 17.5 > Mdn_GPS_ = 6) and risk of psychosis (Mdn_IBA_ = 12.88 > Mdn_GPS_ = 8.5) were higher for the IBA group than for the GPS group. 

The results from the analysis of K-SADS modules are presented in Table 3. 

SOFAS turned out to be significant (*p* < 0.001) when explaining the differences between the examined groups, which indicates a dissimilarity in social and occupational functioning (Table 1).

## 4. Discussion

Consistent with previous studies [52,53], we found that the levels of anxiety, suicide risk, and psychache were similar in both the IBA and SAA groups, and therefore worth considering in general suicide risk assessment. Interestingly, although we observed depressive symptoms in both the SAA and IBA groups, their intensity was significantly higher within the SAA group. Therefore, it seems that adolescents with a higher level of depression symptoms are particularly at risk of suicide attempts. We also observed clear differences between the analyzed groups of patients at the level of psychotic symptoms. The mean score on the PQ-16 for the SAA group did not result in either a schizophrenia diagnosis or verification that it will develop, but analysis of the K-SADS data provided us with some evidence that vocal hallucinations and delusions may result in the predominance of prodromal symptoms among adolescents committing suicide (as observed in the SAA group). These results confirmed the hypothesis that psychotic symptoms were stronger within the SAA group and that adolescents acting on suicidal ideation with a strong desire to die are at greater risk of future suicidal attempts. This strongly suggests that adolescents with psychosis symptoms should be treated as being at a greater risk of suicide than other groups.

Additionally, SOFAS data suggested differences in social relatedness between the examined groups, which suggests a relationship between suicidal risk and social functioning in adolescents. A strong connection between poor relationships and suicidal ideation was found in previous research [54,55,56]. Moreover, a negative association between the attitudes toward school, school performance, and suicide was found in the work of Juon, Nam, and Ensminger [57], as well as in Buddeberg et al. [58]. The possible applications of this research can be addressed to both the primary and secondary prevention of suicide, highlighting the need for careful suicide risk evaluation as part of hospital discharge protocol for adolescents after a suicide attempt and hospitalization related to instrumental suicidal behavior [59]. However, the obtained results were based on a small sample size, which constituted a limitation of the study. In addition, the dispersion of age between the groups was quite big, with a range of 6 years (SD = 1.5 years; σ² = 2.39), making the studied sample heterogenic. Due to the preliminary nature of the study, future research could increase the sample size, focusing on clarifying the diagnostic criteria of different SRBs, which could then delineate the etiological factors associated with suicide attempts and instrumental SRB.

## 5. Conclusions 

Adolescent pediatric patients that were grouped according to either undertaking a non-fatal suicide attempt (SAA), displaying instrumental suicide-related behavior (IBA), or the general population (GPS) differed in terms of several important clinical characteristics. In our study, we found that (a) depressive symptoms were present in both the SAA and IBA groups but their intensity was significantly higher within the SAA group; (b) trait anxiety and state anxiety appeared to be higher for both IBA and SAA groups than for the GPS group, with no significant difference between the IBA and SAA groups; (c) the SAA group was characterized by a significantly higher level of psychotic symptoms than the IBA group; (d) the SAA group had a significantly higher risk of suicide than the IBA group; and (e) no differences in psychache level were observed between the SAA group and the IBA group.

## Figures and Tables

**Table 1 ijerph-17-05568-t001:** Sociodemographic and clinical group characteristics (N = 36).

Characteristics		SAA (n = 12)		IBA (n = 12)		GPS (n = 12)	χ²	*p* (Fisher’s Test)	ANOVA (F; *p*)
Age	15.833 ± 1.586		15.083 ± 1.505		14.667 ± 1.435		-	-	(1.839; 0.175)
Female gender	5	(42%)	9	(75%)	5	(42%)	3.567	0.202	-
SOFAS	42.583 ± 12.139		62.5 ± 16.026		91.091 ± 5.647		-	-	(45.639; <0.001)
Previous hospitalizations	10 (83%)		10 (83%)		1	(8%)	23.061	<0.001	-
Serious injuries	1	(8%)	0	(0%)	0	(0%)	2.473	0.294	-
Head injuries	2	(17%)	1	(8%)	0	(0%)	2.718	0.274	-
Other current or past significant medical health problems	3	(25%)	1	(8%)	0	(0%)	4.939	0.088	-
Problems with social relatedness	9	(75%)	3	(25%)	0	(0%)	20.207	<0.001	-

SAA: Suicidal Act Adolescents Group; IBA: Instrumental Behavior Adolescents Group; GPS: General Population Sample; SOFAS: Social and Occupational Assessment Scale (range 0–100).

**Table 2 ijerph-17-05568-t002:** Mean scores for STAI-trait, STAI-state, TPS, and CES-DC for the different groups.

Group	STAI-Trait		STAI-State		TPS		CES-DC	
	*M*	*SD*	*M*	*SD*	*M*	*SD*	*M*	*SD*
GPS	30.25	5.413	35.25	7.02	17.667	4.097	32.67	2.61
IBA	51.583	10.004	56.75	7.96	36.083	4.542	47.42	11.07
SAA	55.5	8.806	60.67	9.06	41.181	6.242	56.08	3.03

STAI: The State–Trait Anxiety Inventory; TPS: The Psychache Scale; CES-DC: the Center of Epidemiological Studies Depression Scale for Children; GPS: General Population Sample; IBA: Instrumental Behavior Adolescents Group; SAA: Suicidal Act Adolescents Group.

**Table 3 ijerph-17-05568-t003:** Group comparisons for the Kiddie Schedule for Affective Disorders and Schizophrenia (K-SADS) modules.

Module	SAA	SAA (%)	IBA	IBA (%)	GPS	GPS (%)	Three-Group Comparison (U; *p*)	η²	Post Hoc Analysis (MW U-Test)	1 vs. 2 (U; *p*)	η²	2 vs. 3 (U; *p*)	η²
Depression Mood (Threshold)	4	33%	5	42%	0	0%	3.290; 0.193	-	(1 = 2; 2 = 3)	-		-	-
Depression Mood (Threshold) in the Past	2	17%	1	8%	0	0%	0.237; 0.888	-	(1 = 2; 2 = 3)	-		-	-
Irritability and Anger (Threshold)	5	42%	3	25%	0	0%	**8.785; 0.012**	0.206	(1 > 2; 2 = 3)	(44.5; 0.049)	0.105	-	-
Irritability and Anger (Threshold) in the Past	2	17%	0	0%	0	0%	3.318; 0.190	-	(1 = 2; 2 = 3)	-	-	-	-
Anhedonia, Lack of Interest, Apathy, Low Motivation or Boredom (Threshold)	4	33%	6	50%	0	0%	**7.987; 0.018**	0.181	(1 = 2; 2 = 3)	-	-	-	-
Anhedonia, Lack of Interest, Apathy, Low Motivation or Boredom (Threshold) in the Past	1	8%	0	0%	0	0%	0.937; 0.626	-	(1 = 2; 2 = 3)	-	-	-	-
Recurrent Thoughts of Death (Threshold)	4	33%	0	0%	0	0%	**22.580;** **<0.001**	0.624	(1 > 2; 2 = 3)	(16; p < 0.001)	0.436	-	-
Recurrent Thoughts of Death (Threshold) in the Past	1	8%	0	0%	0	0%	4.891; 0.087	-	(1 = 2; 2 = 3)	-	-	-	-
Suicidal Ideation (Threshold)	7	58%	1	8%	0	0%	**20.972;** **<0.001**	0.575	(1 > 2; 2 > 3)	(28.5; 0.004)	0.263	(29; 0.005)	0.251
Suicidal Ideation (Threshold) in the Past	0	0%	1	8%	0	0%	0.331; 0.847	-	(1 = 2; 2 = 3)	-		-	-
Suicidal Acts—Intent (Threshold)	10	83%	8	67%	0	0%	**25.100;** **<0.001**	0.7	(1 = 2; 2 > 3)	-	-	(8; <0.001)	0.569
Suicidal Acts—Intent (Threshold) in the Past	1	8%	2	17%	0	0%	0.075; 0.963	-	(1 = 2; 2 = 3)	-	-	-	-
Suicidal Acts—Medical Lethality (Threshold)	10	83%	5	42%	0	0%	**28.210;** **<0.001**	0.794	(1 > 2; 2 > 3)	(42; 0.045)	0.125	(0.0; <0.001)	0.72
Suicidal Acts—Medical Lethality (Threshold) in the Past	2	17%	0	0%	0	0%	**6.541; 0.038**	0.138	(1 > 2; 2 < 3)	(39; 0.049)	0.151	(36; 0.007)	0.18
Non-Suicidal, Self-Injurious Behavior (Threshold)	8	67%	3	25%	0	0%	**16.802; <0.001**	0.449	(1 > 2; 2 > 3)	(35; 0.015)	0.19	(48; 0.047)	0.08
Non-Suicidal, Self-Injurious Behavior (Threshold) in the Past	1	8%	0	0%	0	0%	1.449; 0.485	-	(1 = 2; 2 = 3)	-	-	-	-
Hallucinations—Voices (Threshold) Child	4	33%	0	0%	0	0%	**22.863; <0.001**	0.632	(1 > 2; 2 = 3)	(18; <0.001)	0.405	-	-
Hallucinations—Visions (Threshold) Child	2	17%	0	0%	0	0%	4.912; 0.086	-	(1 = 2; 2 = 3)	-	-	-	-
Delusions (Threshold)	1	8%	0	0%	0	0%	**16.453; <0.001**	0.438	(1 > 2; 2 = 3)	(29.5; 0.004)	0.251	-	-
The Suicide Behaviors Questionnaire–Revised (SBQ-R)							**31.481; <0.001**	0.893	(1 > 2; 2 > 3)	(0; <0.001)	0.72	(0; <0.001)	0.717
Prodromal Questionnaire (PQ-16)							**24.244; <0.001**	0.674	(1 > 2; 2 > 3)	(0; <0.001)	0.72	(31.5;0.048)	0.122

SAA: Suicidal Acts Adolescent Group; IBA: Instrumental Behavior Adolescent Group; GPS: General Population Sample; MW: Mann–Whitney. Bold print indicates significant results.

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
