# Peer review of "Clinical and Psychosocial Characteristics of Adolescent Pediatric Patients Hospitalized after Different Types of Suicidal Behaviors—A Preliminary Study"

_ijerph, 2020, doi:10.3390/ijerph17155568_

Round 1

Reviewer 1 Report

This manuscript tackles the sensitive area of youth suicide and attempts to identify factors that can be used to differentiate between youth who make suicide attempts that are deemed to be have the intention of success and others who make attempts that don't appear to be intended to be successful.

I have some comments as below

  1. The case definition between SAA and ISRB groups is determined by clinical psychologist and paediatrician.  How did they resolve disagreement?  How confident can we be that these classifications are accurate?  Is there a standardised criteria?  The manuscript mentions characteristics for consideration but is there a more tight definition to apply?
  2. Are you able to describe any demographic information about those who refused to participate. Age, gender etc.
  3. Suggest that the presentation of the data can be improved to assist the reader.  I don't think that Figure 1 works very well. It might be my preference but I think these data displayed in a table will be more helpful to the reader.
  4. Tables.  These can be divided up a little more and then the accompanying text related to that specific table is closely located.  I can imagine that a clinician interested in your paper would want to go to the table that has the specific instruments and data
  5. Do the authors have statistical power to undertake the multiple comparison which they report.  With approximately 12 in each group and the range of statistical tests, I am a bit concerned about power and some discussion of this will assist the reader.
  6. There are lots of acronyms and sometimes these are not consistent.  Please consider spelling out in full at various times in the manuscript to assist the reader.  A regular reminder of what these means helps to not have to stop reading and go back to the first occurrence again.
  7. ln 40 7% of 16% or of Polish youth?
  8. ln 56-60 this sentence seems repetitive and needs explaining
  9. Check SA or SAA.  I think they are the same thing.
  10. Check ISRB or IBA.  I think they are the same thing.
  11. 2.1 participants.  It seems out of place to present results in this section. I expect to understand criteria for participant selection and then find the supporting data in results.
  12. Results Figure 1. appear to have very tight confidence intervals?
  13. Ln 329 333 refer to IBS not ISRB?

Reviewer 2 Report

The paper entitled "Clinical and psychosocial characteristics of adolescent pediatric patients hospitalized after different types of suicidal behaviors –
preliminary data"  presents interesting results which, apart from broadening the so-far knoweldge about underpinnings and psychosocial and clinical characteristis of adolescent's sucidal behaviors, might be used in preventive actions adressed to families of adolescents at risk of suicide (and adolescents as well). Before publication, I would suggest to make some amendments, which I put below:

  1. In Abstract section Author/Authors write about differences between groups in SBQ-R and PQ-16 (can you write exactly what were these differences?)
  2. Introduction - suicidal behaviors may have many functions in the period of adolescence. It may be regarded as one of risk behaviors, tipical for adolescence, which stems from neurobiological development of adolescent brain. I think a short information about that in this section of the article, would be reasonable.
  3. Theoretical background is preparred very carefully. However, could you please give some Polish contemporary studies concerning this field of study, if there is such research? (as you write that "only a few studies investigated clinical characteristics of adolescent patients after different types of SRBs"). Maybe there is a specificity of such behaviors in Polish population of adolescents? 
  4. Lines 164-168 - maybe it would be more logical to put these information in Result section?
  5. In table 1 - dots instead of commas; N=36 instead of n=36 (while you refer to all participants).  Statistical symbols such as p, M, SD should be written in italics
  6. As an age range of participants is quite big, I would suggest to write about it in discussion section highlighting that it may be a limitation of this study. Adolescence is a very heterogenic period of life, with dynamic developmental changes in e.g. functionning of brain. 

Reviewer 3 Report

This manuscript presents results of study of hospitalized adolescents after either a suicide attempt or an instrumental suicide-related behavior as well as a general population sample of adolescents matched on several variables with the hospitalized groups. In an attempt to determine if predictors of suicide risk might be found, a number of established measures were administered and compared for the groups. Group differences and non-differences were observed and reported that may inform larger scale research and possibly clinical assessment with adolescents.

The strengths of the manuscript include the use of an array of established methods with known psychometric properties of reliability and validity. The inclusion of a general population sample (assessed to be without any psychiatric disorders) is another strength as is the study of not just suicide attempters but also those with instrumental suicide-related behaviors.

There are some issues that might be addressed.

  • Sample Size: This report is labeled a preliminary data study and the small sample (n = 36, 12 in each of 3 groups) is briefly noted as an issue in the discussion. However, some clearer statement of caution regarding the preliminary nature of the study and its findings could be included as well as the need for the results to be tested in a full sample of these groups.
  • Participants: From 914 pediatric patients, 15 were "identified" as admitted following a suicide attempt and 23 for instrumental suicide-related behaviors. Of these 38, however, 3 and 11, respectively did not agree to participate in the study. In this total of 914 pediatric patients, were these 38 the entire number that had been hospitalized for these reasons? Based on hospital records, were the 14 who declined to participate different in any way from the 24 who were ultimately studied (e.g., age, gender, etc.)? With respect to the 12 adolescents "recruited" from schools, how exactly were these students recruited and identified? It is stated that they were age- and gender-matched to the SAA and IBA groups; since there were 12 in each of these other groups, to whom among the two groups were the 12 matched since there were 24 in the SRB groups total? That is, how exactly was the matching done if the SAA and IBA groups were not already "matching" on gender and age (Table 1 indicates that they were not both comprised of the same number of adolescent boys and girls; was the control group matching conducted just for the SAA group?)? A bit more explanation seems needed. A final thought relates to the total group of pediatric patients (n = 914). Were the remaining over 800 patients all hospitalized for other mental health conditions? If not, a second matched sample might have been included from among that group (assuming that they were hospitalized for non-psychiatric conditions). If the facility is a psychiatric setting, then this may not be an appropriate group with which to compare.
  • Other Points: (1) on page 3 of 19, line 77 the term "commit suicide" is utilized. Current practice in the field is to avoid such terminology and instead using something like "died by suicide" or even "suicided." (2) Very minor, but on the same page (3 of 19) on line 93 the author "Shneidman" is misspelled "Schneidman." The spelling is correct in the reference list but not on this text page. (3) The word "data" is plural and should be used with a plural verb, so that on page 3 of 19, line 83 and 13 of 29 on line 321 "data supports" and "data suggests" should be "data support" and "data suggest." Similarly, in line 169 on page 5 of 19 "Both written and informed consent was…" should be "Both written and informed consent were…".

Round 2

Reviewer 1 Report

Thank you for considering my review and for the resultant changes to the manuscript.